# Urban Land Use and Land Cover Change Analysis Using Random Forest Classification of Landsat Time Series

Saeid Amini [1], Mohsen Saber [2], Hamidreza Rabiei-Dastjerdi [3] and Saeid Homayouni [4,*]

1   Department of Geomatics Engineering, Faculty of Civil Engineering, University of Isfahan, Isfahan 81746-73441, Iran; s.amini@eng.ui.ac.ir
2   Department of Geospatial Information Engineering, Faculty of Surveying and Geomatics, University of Tehran, Tehran 14174-66191, Iran; m.saber@ut.ac.ir
3   School of Computer Science and CeADAR, University College Dublin (UCD), D04 V1W8 Dublin, Ireland; hamid.rabiei@ucd.ie
4   Centre Eau Terre Environnement, Institut National de la Recherche Scientifique, Quebec City, QC G1K 9A9, Canada
*   Correspondence: saeid.homayouni@inrs.ca

**Abstract:** Efficient implementation of remote sensing image classification can facilitate the extraction of spatiotemporal information for land use and land cover (LULC) classification. Mapping LULC change can pave the way to investigate the impacts of different socioeconomic and environmental factors on the Earth's surface. This study presents an algorithm that uses Landsat time-series data to analyze LULC change. We applied the Random Forest (RF) classifier, a robust classification method, in the Google Earth Engine (GEE) using imagery from Landsat 5, 7, and 8 as inputs for the 1985 to 2019 period. We also explored the performance of the pan-sharpening algorithm on Landsat bands besides the impact of different image compositions to produce a high-quality LULC map. We used a statistical pan-sharpening algorithm to increase multispectral Landsat bands' (Landsat 7–9) spatial resolution from 30 m to 15 m. In addition, we checked the impact of different image compositions based on several spectral indices and other auxiliary data such as digital elevation model (DEM) and land surface temperature (LST) on final classification accuracy based on several spectral indices and other auxiliary data on final classification accuracy. We compared the classification result of our proposed method and the Copernicus Global Land Cover Layers (CGLCL) map to verify the algorithm. The results show that: (1) Using pan-sharpened top-of-atmosphere (TOA) Landsat products can produce more accurate results for classification instead of using surface reflectance (SR) alone; (2) LST and DEM are essential features in classification, and using them can increase final accuracy; (3) the proposed algorithm produced higher accuracy (94.438% overall accuracy (OA), 0.93 for Kappa, and 0.93 for F1-score) than CGLCL map (84.4% OA, 0.79 for Kappa, and 0.50 for F1-score) in 2019; (4) the total agreement between the classification results and the test data exceeds 90% (93.37–97.6%), 0.9 (0.91–0.96), and 0.85 (0.86–0.95) for OA, Kappa values, and F1-score, respectively, which is acceptable in both overall and Kappa accuracy. Moreover, we provide a code repository that allows classifying Landsat 4, 5, 7, and 8 within GEE. This method can be quickly and easily applied to other regions of interest for LULC mapping.

**Keywords:** cloud computing; Google Earth Engine; Landsat time series; LULC change detection; pan-sharpening; pixel-based image classification; Random Forest

## 1. Introduction

Since the 1950s, the world's population has grown from 2.6 billion to 7.7 billion, and is expected to reach approximately 9.7 billion by 2050 [1]. A larger population causes an increased demand for energy, food, housing, water, transportation, and healthcare. To provide these demands, humans have exploited natural resources and consequently changed the surface of the Earth. Land use and land cover (LULC) change is the most obvious

indicator of the Earth's surface changes [2,3]. Because of communities' social and physical characteristics, the distribution of LULC changes varies in space and time. According to recent studies, LULC change can show the adverse effects of different environmental and socioeconomic factors on the Earth's surface, e.g., climate, water balance, biodiversity, and terrestrial ecosystems [4–6]. Among these factors, the impact of the LULC on the ecosystem has become the most attractive topic among researchers [3,7–10]. During past decades, remote sensing (RS) systems have collected valuable information from Earth's surfaces. Although RS provides valuable data for researchers, standard software packages are not fully functional for delivering a comprehensive solution for analyzing and managing Earth data [10,11]. During past decades, a variety of RS sensors, such as Moderate Resolution Imaging Spectroradiometer (MODIS), Satellite Pour l'Observation de la Terre (SPOT), Landsat, and Sentinel 2 have been launched [12,13]. Different methods for extracting information from these remotely sensed data have been introduced and developed. Among these data, Landsat is the only sensor providing data for more than four decades and still is active. The spatial resolution of this dataset provides the possibility to study Earth's surface dynamics which is a necessity for policy development, management, and scientific inquiries [1]. Landsat dataset provides almost high spatial (30 m and 15 m in PAN band in Landsat 7 and 8) and temporal (8 days, if we combine data from both satellites and 16 days with a single satellite) data to a large extent [3,13]. Other sensors such as Sentinel-2 cannot provide a continuous data source for this long period (4 decades) or suffer from other problems such as lower spatial, spectral, or temporal resolution.

There are many classification algorithms for LULC mapping, including parallelepiped [12,14], minimum distance (MD) [14], maximum likelihood (ML) [15], fuzzy classification (FC) [16], Artificial Neural Network (ANN) [17], Support Vector Machine (SVM) [14], Random Forest (RF) [3,18,19], deep learning (DL) [20,21], and deep transfer learning (DTL) [21,22]. Among these classifiers, RF, due to its better efficiency and higher accuracy, relatively low computational cost, and the need for a few parameters, has gained great popularity and become one of the best candidates in LULC mapping [17,21]. Although DL and DTL methods have become an effective computational approach in machine learning in recent years, these methods require extensive data and limit complex computations in the cloud platform [20].

In recent years, to overcome the challenge of big data analytics in RS, Google developed the Google Earth Engine (GEE) platform to process a huge amount of data for a long time [23]. The GEE project was launched in 2010 and has become the most popular big Earth Observations (EO) analysis platform [10,24]. GEE encompasses many datasets comprising raw and preprocessed datasets and elevation models. GEE also covers various regional, national, and global extents [10]. Among the available datasets in GEE, the Landsat imagery archive (e.g., Landsat 1–3 (1972–1983), Landsat 4 (1982–1993), Landsat 5 (1984–2012), Landsat 7 (1999–present), and Landsat 8 (2013–present)) is one of the most commonly used products for various agro-environmental applications such as multi-temporal image classifications [25], multi-temporal cloud masking [26], multi-temporal settlement and population mapping [27], etc. The GEE platform also decreases the time and effort dedicated to preprocessing stages applied to satellite image data available in this environment [1].

Various approaches for LULC change analysis are used [27–29]. Researchers usually use time-series indices derived from multispectral satellite data for LULC change analysis. Change analysis based on the spatiotemporal variation of land surface biophysical properties is one of the most commonly used methods. Noi Phan et al. investigated the role of image composition in LULC classification using RF in the GEE platform [17]. This study showed that input features are critical and directly influence final classification accuracy. Mugiraneza et al. examined gray-level co-occurrence matrix (GLCM) texture features with standard spectral features to monitor urban LULC change with Landsat time-series imagery [29,30]. The features in LULC classification are generally categorized into spectral, textural, and geometrical features. The type of dataset, area conditions, and auxiliary data availability are the main factors behind selecting a type or a combination of these

features in the studies [3,31,32]. Different studies have shown the relation between land surface temperature (LST) and LULC [33–35], and it is worth checking the magnitude of this feature's impact in Landsat LULC classification using calculated variable importance (VI) in RF. Gilbertson et al. [34] also showed that pan-sharpening Landsat 8 (L8) imagery for classifying agricultural fields is beneficial for LULC classification, and final results would become more accurate. Gilbertson et al. applied a pan-sharpening process to L8 multi-temporal imagery for differentiating between crop types using different classification techniques [34].

This study differs from previous efforts to map LULC in three aspects. First, we used a pan-sharpening method to increase the spatial resolution of RGB bands in top-of-atmosphere (TOA) Landsat data for Landsat 7–8 data. Researchers use surface reflectance (SR) instead of TOA Landsat data [1]. Given that the panchromatic band is not available in SR Landsat data, it prevents researchers from using the pan-sharpening method. In the TOA product of Landsat, a higher level of radiometric corrections is at the expense of losing higher spatial resolution. Second, we used VI in RF to determine essential features in the LULC mapping process. The results showed that LST and DEM are valuable features in LULC mapping. The increasing use of cloud platforms provided more features in the classification process and allowed us to determine suitable features in classification and change detection applications. Selection of essential variables is mandatory for data sets with many variables. Third, we compared the results of our proposed method with the Copernicus Global Land Cover Layers (CGLCL) map to prove the efficiency of our proposed method. The CGLCL is a global discrete land cover map at 100 m resolution for processing Earth Observation (EO) data. This map is generated by using several proven individual methodologies applied to high-quality external data (e.g., PROBA-V, Sentinel, Landsat, DEM, and land-sea mask) [35]. This framework increases the classification accuracy for seven to nine generations of the Landsat series, enabling continuous and long-term monitoring and analysis of LULC change. The rest of the paper is organized as follows. First, the dataset used in this study is described. Then, the methodologies of the RF algorithm for LULC classification in the GEE platform are presented in Section 2. In Section 3, the results and analyses are presented. A discussion is presented in Section 4, and finally, the paper's findings are summarized in Section 5.

## 2. Materials and Methods

### 2.1. Study Area

Isfahan is the capital city of Isfahan province in Iran at an elevation of about 1600 m, roughly 340 km south of the capital city of Tehran. Isfahan is one of Iran's largest cities, located at 32°38′41″N and 51°40′03″E (Figure 1). Isfahan has experienced noticeable LULC change during the past decades, especially in urban growth [34,35]. The metropolitan area of Isfahan covers a total area of 551 km$^2$, with almost 2 million inhabitants. Its comprehensive economic, historical, and geographical strength ranks this city among the top three cities in Iran. We demonstrate the effectiveness of our methodology in the Isfahan city region, including the Isfahan metropolitan area and several surrounding smaller cities (in total ~42 cities with a population of approximately 3,400,000 [36,37]).

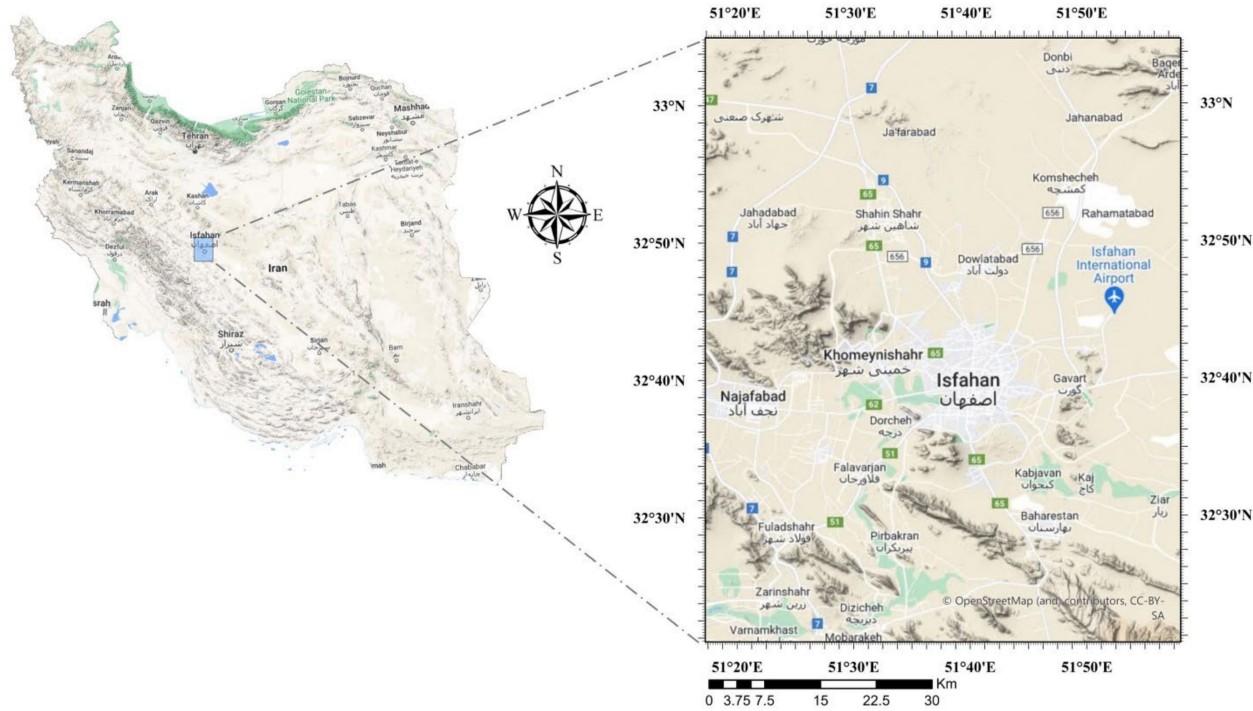

**Figure 1.** Study area.

*2.2. Earth Observations*

This study used a CGLCL map and some auxiliary data to extract LULC from Landsat collection data from 1985 to 2019. Table 1 provides more details about these datasets.

**Table 1.** Description of datasets included in the study.

| Name | Dataset Provider | Dataset Availability | Resolution |
|------|-----------------|---------------------|------------|
| USGS Landsat 5 TM Collection 1 Tier 2 TOA Reflectance | USGS/Google | 1984-03-05–2012-04-26 | 30 m |
| USGS Landsat 7 Collection 1 Tier 2 TOA Reflectance | USGS/Google | 1999-05-28–2021-12-31 | 15–30 m |
| USGS Landsat 8 Collection 1 Tier 2 TOA Reflectance | USGS/Google | 2013-03-18–2022-01-02 | 15–30 m |
| NASA SRTM Digital Elevation 30 m | NASA/USGS/JPL-Caltech | 2000-02-11–2000-02-22 | 30 m |
| CGLS-LC100 collection 3 | Copernicus | 2015-01-01–2019-12-31 | 100 m |
| USGS Landsat 8 Level 2, Collection 2, Tier 1 (Surface Reflectance) | USGS | 2013-03-18–2022-05-23 | 30 m |

We also combined the digital elevation model (DEM), slope and aspect derived from the digital terrain model (DTM), and multispectral bands for LULC classification. We used the Shuttle Radar Topographic Mission (SRTM) data for one arc-second global product (around a 30-m spatial resolution) to check the importance of height as an input feature in the classification procedure. More details about DTM specifications can be found at https://www2.jpl.nasa.gov/srtm/ (accessed on 26 April 2022).

2.2.1. Landsat Imagery

The Landsat program has been used since 1972 in research and applications in spatial planning, education, agriculture, geology, forestry, and surveillance [36,38]. Since 2008, due to removing the monetary cost of accessing its data, it has become one of the most known and publicly available remotely sensed data. Landsat downloaded images reached about 20 million in 2017 [39,40]. The United States Geological Survey (USGS) provides

Landsat data, and users can freely access Landsat 4–8 on GEE. GEE covers a variety of Landsat data processing methods. These methods are mainly categorized into at-sensor spectral radiance, TOA reflectance, and SR [40]. For each satellite, USGS products based on quality are grouped into Real-Time (RT), Tier 2 (T2), and Tier 1 (T1). T1 is the most accurate product and is the best candidate for time series analysis processes [41]. Thematic Mapper (TM) sensor, Enhanced Thematic Mapper Plus (ETM+; an improved version of TM), and Operational Land Imager (OLI) acquire data in the visible and infrared spectral regions in Landsat-4 and Landsat-5, Landsat 7, and Landsat 8, respectively. This paper used six cloud-free Landsat images from satellites 5 to 8 between 1985 and 2019. Table 2 describes the properties of the spectral bands in Landsat collections used in this study.

**Table 2.** Proposed LULC classification scheme and the corresponding description.

| Code | LULC Type | Abbreviation | Description |
|---|---|---|---|
| 20 | Shrubs | Shrub | Woody perennial plants with persistent and woody stems and no defined main stem are less than 5 m tall. The shrub foliage can be either evergreen or deciduous. |
| 40 | Cultivated and managed vegetation/agriculture | Cult | Temporary crops cover lands after harvest and a bare soil period (e.g., single and multiple cropping systems). Perennial woody crops will be classified as the appropriate forest or shrub LC type. |
| 50 | Urban/built up | Urb | Buildings and other human-made structures cover LC. |
| 60 | Bare/sparse vegetation | Bare | Land with exposed soil, sand, or rock never has more than 10% vegetative cover at any time of the year. |
| 80 | Permanent water bodies | Wat | Lakes, reservoirs, and rivers. It can be either fresh or salt-water bodies. |
| 90 | Herbaceous wetland | Wet | Lands with a permanent mixture of water and herbaceous or woody vegetation. The vegetation can be present in either salt, brackish, or freshwater. |
| 125 | Open forest, mixed | For_Mix | Open forest, mixed. |
| 126 | Open forest, not matching | Op_For_NM | The Open Forest does not match any of the other definitions. |

The first, second, and third Landsat 5 Thematic Mapper composite (LANDSAT/LT05/C01/T1_TOA Path: 164; Row: 37) was constrained between 1 July and 31 August 1985, 1993, and 2008, respectively. The Landsat 7 Enhanced Thematic Mapper composite (L7-ETM+) was selected between 1 July and 31 August 2000 (LANDSAT/LT07/C01/T1_TOA). The 2013 and 2019 images were acquired using the Landsat 8 Operation Landsat Imager (L8-OLI) sensor with LANDSAT/LC08/C01/T1_TOA identification code. An annual cloud-masked composite was collected by constraining the acquisition between 1 July and 31 August 2013 and 2019 because of the low vegetation and cloud cover availability.

### 2.2.2. CGLCL

In July 2017, the first Copernicus Global Land Service Land Cover Map at 100 m (CGLS-LC100) was released. This product mapped the African continent in the 2015 reference year. It combined several ancillary datasets and a high-quality LULC reference sites database. It was derived from the vegetation instrument's 100 m time series onboard the PROBA satellite (PROBA-V) [41–43]. The successful experience of the first collection of CGLS-LC100 maps, including high accuracy and good alignment with other continental LULC maps, results in the second LULC map release. The second collection of CGLS LULC maps was released in May 2019 and showed around 6% better accuracy (about 80%) [41,44]. For a more detailed explanation, including detailed technical descriptions of algorithms, we refer to the Algorithm Theoretical Basis Document (ATBD) [44]. The third collection of CGLS LULC maps is available from January 2015 to January 2020. This collection consists of 23 classes, and the spatial resolution of this dataset is 100 m. This collection is described in Table 2.

### 2.2.3. Training and Validation Data

In this study, eight LULC types dominate the investigation area and are fully described in Table 2. The training and validation samples collection was based on visual interpretation of high-resolution images from Google Earth and the CGLCL map. This method is extensively applied and reported in the literature [18,36,45]. For more accurate validation, the training and validation data were separately selected. Table 3 displays more details about the training and validation data according to the complexity of the area used in this study.

**Table 3.** The average number of training and validation points for each LULC class.

| Land Cover Type | Number of Training Pixels | Number of Validation Pixels |
|---|---|---|
| Shrubs | 38,724 | 19,484 |
| Cultivated and managed vegetation/agriculture | 16,107 | 9771 |
| Urban/built up | 2910 | 1470 |
| Bare/sparse vegetation | 169 | 75 |
| Permanent water bodies | 2365 | 1977 |
| Herbaceous wetland | 3387 | 2563 |
| Open forest, mixed | 19,300 | 13,511 |
| Open forest, not matching | 576 | 307 |

### 2.3. Image Preprocessing

Figure 2 illustrates the workflow for generating a LULC map. This workflow consists of four parts. (1) Datasets are introduced, and sample data based on available data are generated using a CGLCL map and visual interpretation. (2) A pan-sharpening method for Landsat 7 and 8 is applied. (3) Based on Landsat data, some indices (e.g., NDVI, DEM, and LST) are computed, and (4) RF classification is performed at two stages based on training data. These two steps train the classifier using training data and perform classification on whole images. Finally, validation methods are applied to evaluate the performance of the classifier and the accuracy of the LULC map. All of these steps were performed in the GEE.

### 2.3.1. Pan-Sharpening

Pan-sharpening combines the spectral information of lower resolution multispectral bands with a panchromatic band's relatively higher spatial resolution over the same area [46,47]. There are several pan-sharpening methods in the literature. Although losing some of the spectral information from the original multispectral information during pansharpening is inevitable, some algorithms attempt to maximize spectral preservation [34]. Most pan-sharpening methods take advantage of the panchromatic image's high-frequency information and use different models to inject this extracted data into resampled multispectral bands [47]. The Intensity, Hue, Saturation (IHS) method is one of the popular methods for pan sharpening. This method works based on conversion between RGB and IHS color space, resulting in higher spatial resolution for multispectral bands [48].

### 2.3.2. Feature Generation

During the last five decades, RS scientists have introduced and developed quantitative and qualitative vegetation indices for evaluating vegetative covers. Such indices are generated based on spectral measurements [48,49].

In this study, two types of spectral-temporal features were used. These features are raw Landsat collection bands and extracted features from raw and auxiliary data (e.g., DEM, slope, and aspect). Details about extracted features from Landsat collection to increase the accuracy of LULC classification are mentioned in Table 4.

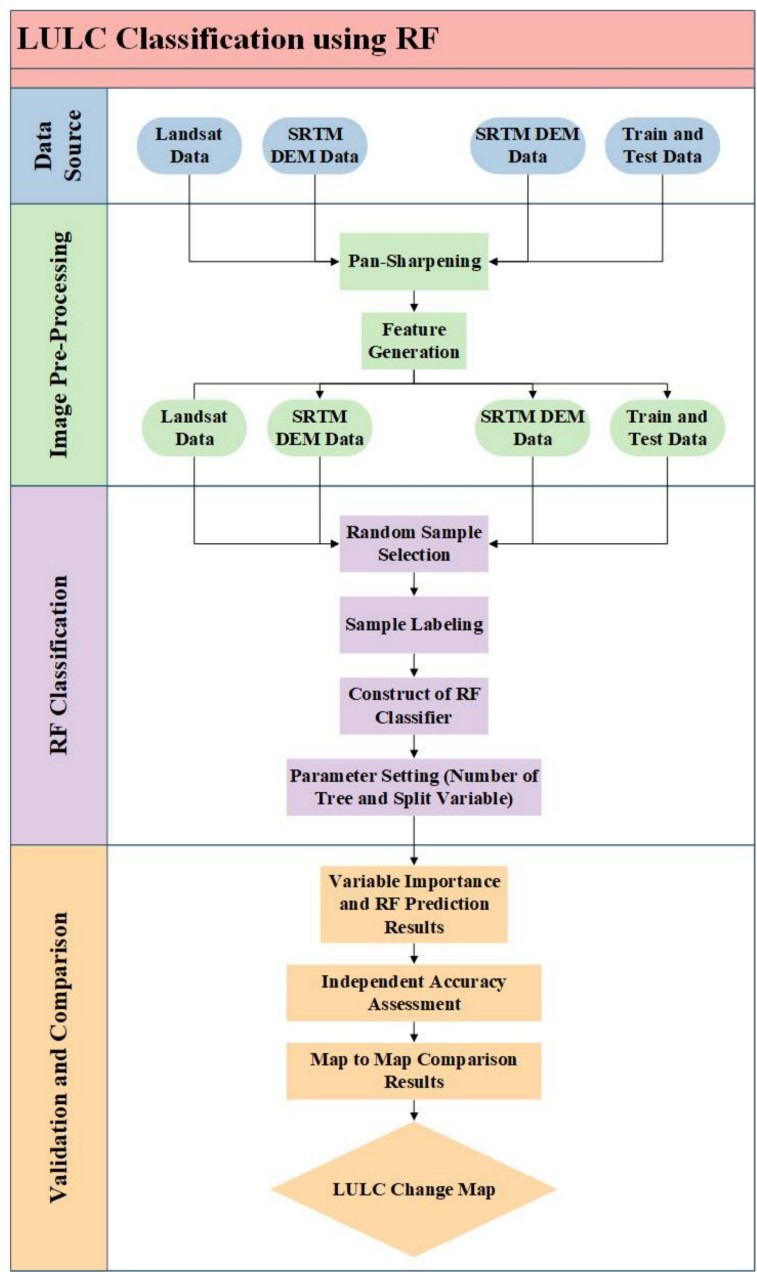

**Figure 2.** The workflow of LULC classification is based on the RF.

**Table 4.** Extracted features from Landsat collection.

| Full Name | Abbreviation | References |
|---|---|---|
| Built-up Area Extraction Method | BAEM | Bhatti et al. [49] |
| Built-Up Area | BU | Kaimaris et al. [50] |
| Dry Built-up Index | DBI | Rasul et al. [51] |
| Dry Bare-Soil Index | DBSI | Rasul et al. [51] |
| Enhanced Built-Up and Bareness Index | EBBI | As-syakur et al. [52] |
| Enhanced Vegetation Index | EVI | Liu & Huete [53] |
| Green-Red Vegetation Index | GRVI | Xu et al. [54] |
| Index-based Built-up Index | IBI | Xu [55] |
| Land Surface Temperature | LST | Sobrino et al. [56] |
| Modified Normalized Difference Water Index | MNDWI | Epting et al. [57] |
| Modified Soil-adjusted Vegetation Index | MSAVI | Storey et al. [58] |
| Normalized Burn Ratio | NBR | Zhao et al. [59] |

**Table 4.** *Cont.*

| Full Name | Abbreviation | References |
|---|---|---|
| Normalized Dry Bareness Index | NDBaI | Zha et al. [60] |
| Normalized Difference Built-up Index | NDBI | Herbei et al. [61] |
| Normalized Difference Vegetation Index | NDVI | Jeevalakshmi et al. [62] |
| Normalized Difference Moisture Index | NDMI | Gao [63] |
| Normalized Difference Water Index | NDWI | McFeeters [64] |
| Soil Adjusted Vegetation Index | SAVI | Huete [65] |
| Simple Ratio Vegetation Index | SR | Birth & McVey [66] |
| Urban Index | UI | Kawamura [67] |
| Visible Green-Based Built-up Indices | VgNIR_BI | Estoque et al. [68] |
| Visible Red-Based Built-up Indices | VrNIR_BI | Estoque et al. [68] |

LST is defined as the measured radiative skin temperature of the land surface in the direction of the remotely sensed sensor and can be applied in different applications investing land surface conditions (e.g., urban climate, vegetation stress, and urban heat islands) [69]. Among the various studies focusing on LST retrieval using Landsat series data, Ermida et al. implemented the LST retrieval method within the GEE platform [69]. The code is freely available to compute LST for Landsat 4, 5, 7, and 8.

*2.4. Image Classification*

Image classification is a subdomain of computer vision dealing with categorizing and labeling groups of pixels within an image [11,70,71]. Four pixel-based approaches are prominent for LULC classification, including supervised, unsupervised, contextual, and hybrid methods [38,50]. The better results in such techniques rely on the complexity of classes, the number of training data, the number and type of input features, and the classifier method [38,50]. RF is one of the best solutions used widely in classification problems.

RF Classifier

The RF classifier as a machine learning technique is the ensemble learning method. This method was first introduced by Breiman [28]. RF combines several tree predictors [70]. Available RF classifiers in GEE generally use six input parameters, which are the number of trees used in classification, the number of variables used at each node, random seed variable for decision tree construction, minimum leaf population, bagged fraction of the input variables for each decision tree, and out-of-bag mode. When the number of trees increases, the overall classification accuracy increases until convergence begins without overfitting [28]. Overfitting can be eliminated if optimized parameter values are set in the RF classifier. One way to select optimized parameters is using the Out-of-Bag (OOB) outputs. The optimized parameter values are selected based on the RF algorithm using the training samples and checking the classification accuracy using test data [3,19,22].

VI is one of the RF classifier products and measures each variable's impact on the overall prediction performance of the model. Computing the decrease of prediction accuracy resulting from randomly permuting the values of a variable helps calculate VI. The lower the prediction accuracy, the less important a variable is, and vice versa [53,54]. In other words, several features are randomly selected at each node. The most important feature among these features is selected based on criteria such as the Gini index (the more the Gini index decreases for a feature, the more critical it is). The selected feature is used to perform splitting at the node. When the classification is completed using the RF algorithm, the number of iterations of each attribute is counted. A feature that is repeated in more nodes is more important.

*2.5. Accuracy Assessment*

Accuracy assessment is one of the most important final steps for image classification. Accuracy assessment helps to provide a quantitative measure of how effectively pixels (or segments) are assigned to the correct LULC classes. In other words, accuracy is defined as the degree of similarity between the produced and reference maps. Different approaches and indices for accuracy assessment have been introduced and developed in past years. The overall accuracy (OA) and Kappa coefficient are two commonly used overall measures of accuracy [72]. OA shows the probability of correct classification of a randomly selected location on the map [73]. The Kappa coefficient varies between −1 and 1. A value of 0 implies that the classification is similar to a random classification. A negative value for Kappa indicates that the classification is significantly worse than random classification. A value close to 1 shows that the classification is correct [74]. We also used accuracy, F1-Score, misclassification rate, precision, recall, and specificity to evaluate the performance of the proposed method [74,75].

$$\text{Accuracy} = \frac{\text{TN} + \text{TP}}{\text{TN} + \text{TP} + \text{FP} + \text{FN}} \tag{1}$$

$$\text{F1} - \text{Score} = \frac{2 \times \text{Precision} \times \text{Recall}}{\text{Precision} + \text{Recall}} = \frac{2 \times \text{TP}}{2 \times \text{TP} + \text{FP} + \text{FN}} \tag{2}$$

$$\text{Misclassification Rate} = \frac{\text{TP} + \text{FN}}{\text{TP} + \text{TN} + \text{FP} + \text{FN}} = 1 - \text{Accuracy} \tag{3}$$

$$\text{Precision} = \frac{\text{TP}}{\text{TP} + \text{FP}} \tag{4}$$

$$\text{Recall} = \frac{\text{TP}}{\text{TP} + \text{FN}} \tag{5}$$

$$\text{Specificity} = \frac{\text{TN}}{\text{TN} + \text{FP}} \tag{6}$$

where FN, FP, TP, and TN, are the number of false-negative, false positive, true positive, and true negative observations, respectively. First, the metric for each category was calculated, and then the average score across all categories was computed.

## 3. Results

The present study aimed to design a method for reliable LULC classification. As mentioned in Section 2, our methodology consists of four main sections. First, we introduced Landsat images and collected training and test data individually. Next, we introduced standard or pan-sharped spectral bands and additional variables for classification. Then, RF classification was applied to classify images. Finally, we performed an accuracy assessment using test data to evaluate the performance of the results of the LULC map and determined the importance of each input variable using VI in the Isfahan region, a major city in the Middle East.

In this study, we performed two map-to-map comparison approaches and independent accuracy assessments to evaluate the performance of the proposed methodology. To this aim, we used CGLCL data and Landsat SR data.

Pan-sharpening is one of the steps used for Landsat 7 and Landsat 8 data. The pan-sharpening was applied only on data acquired by sensors mounted on Landsat 7 (ETM+) and 8 (OLI) because the pan band is not available in sensors used in Landsat 5 (TM) and an earlier generation of Landsat instruments (MSS). In this step, we used a pan band to increase the spatial accuracy of RGB bands. Figure 3 shows one sample comparing the 15 m pan-sharpened image results with the standard version of 30 m RGB bands in 2019. According to Figure 3, the roads and buildings are more vivid and clear to discriminate, and it would help the classifier extract classes more effectively.

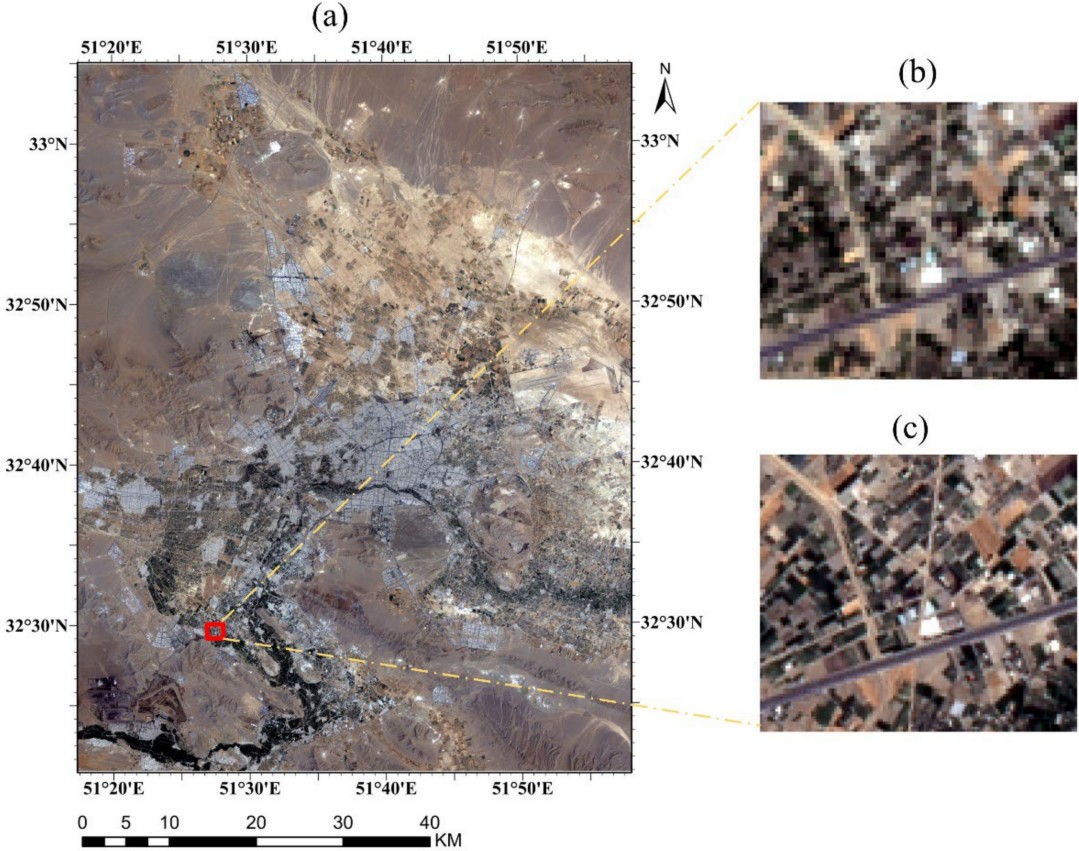

**Figure 3.** Effect of the pan-sharpening on the 2019 image: (**a**) whole area; (**b**) selected area before pan-sharpening; (**c**) selected area after pan-sharpening. (Red rectangle shows sample area for comparison).

This study investigated the relative importance of input features, including Landsat data bands, several known vegetation and building indices, and other auxiliary data for mapping the LULC pattern. For this purpose, we used an RF classifier based on the GEE platform. According to the recommendations of similar studies [18,58,59] and the result of OOB error [18] from our data (Figure 4), we selected 50 trees (ntree = 50). At the same time, the square root of the total number of features (mtry) was set as the default value. Figure 4 suggests that the classifier's performance remains nearly identical with 50 and 100 decision trees. Table 5 provides more details about the values of parameters set in the RF algorithm. Using the created RF, we conducted this classifier in six periods. Figure 5 presents the results of the LULC map for six considered dates from 1985 to 2019.

**Table 5.** Parameters used for training in RF.

| Parameter | Value |
|---|---|
| The number of decision trees to create. | 50 |
| The number of variables per split. | default: square root of the number of variables |
| The minimum number of Leaf Population. Only nodes containing more training sets than this value are created. | default: 1 |
| The fraction of input to bag per tree. | default: 0.5 |
| The maximum number of leaf nodes in each tree. | default: no limit |
| The randomization seeds. | default: 0 |

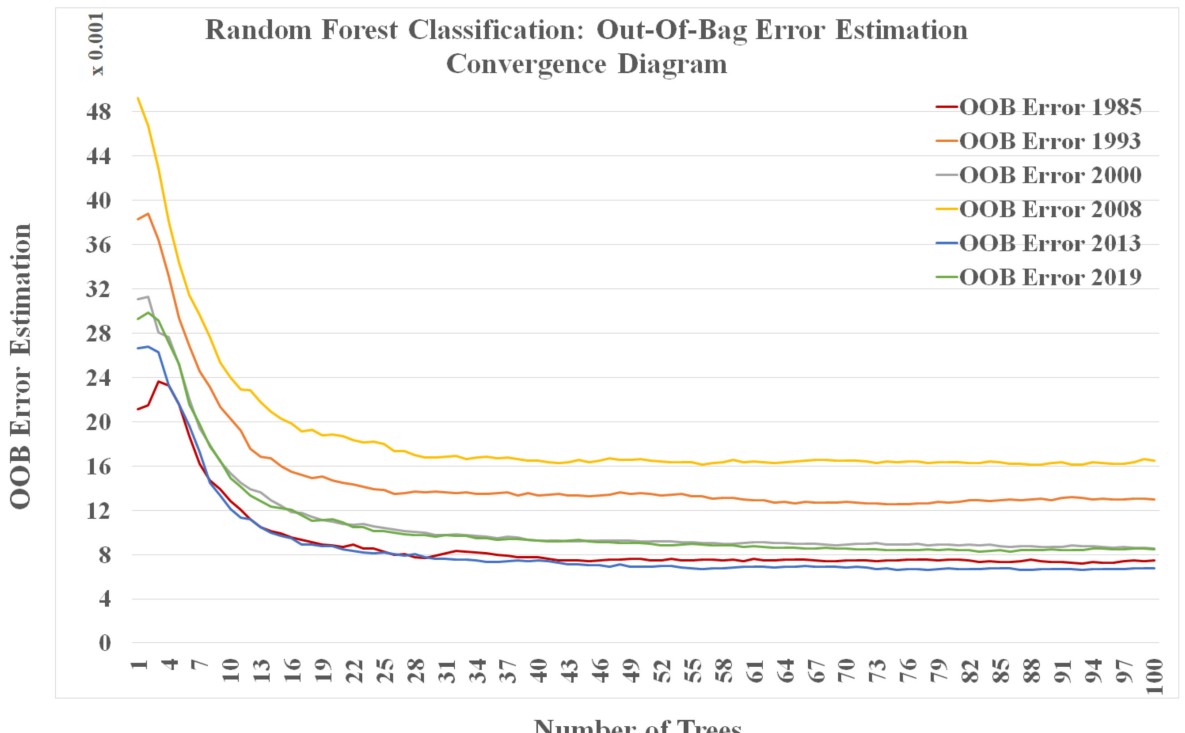

**Figure 4.** Effect of the number of trees of RF on classification accuracy.

After performing RF classification, VI for each input feature was calculated. Figure 6 presents the average feature importance for six proposed periods from 1985 to 2019. We considered 32 features for Landsat 5 time-series data for 1985, 1993, and 2008, 33 features for Landsat 7 time-series data for 2000, and 34 features for Landsat 8 time-series data for 2013–2019. Regarding Landsat 5 time series data, DEM, LST, slope, and B6 were the four most important variables. For Landsat 7 time series data, DEM, LST, NBR2, and B1 and Landsat 8 time series data, DEM, B11, LST, and B10 featured high importance variables. However, RVI, NDVI, and VrNIR_BI only marginally help classify all datasets. The general results also displayed that DEM, LST, B10, B11, and NBR2 are the most important variables, while RVI, VrNIR_BI, NDVI, and MSAVI are less influential in classification processes.

*3.1. Independent Accuracy Assessment*

Producer's accuracies, user's accuracies, OA, Kappa coefficients, accuracy, F1-score, misclassification rate, precision, recall, and specificity for all six periods are listed in Figure 7 and Table 6. In general, all datasets produced high accuracies (OA and Kappa range from 94.176% to 97.554% and 0.908 to 0.962, Table 6). On average, urban, bare land, shrub, and forest not matching area were classified with the highest accuracy, followed by water, urban, and cultivated. The lowest accuracies were observed with forest mixed and wetland classes. These low accuracies are caused by the very low UA and PA of the 2000 and 2008 periods. Figure 7 and Table 6 present the details of LULC classification statistics on six considered dates.

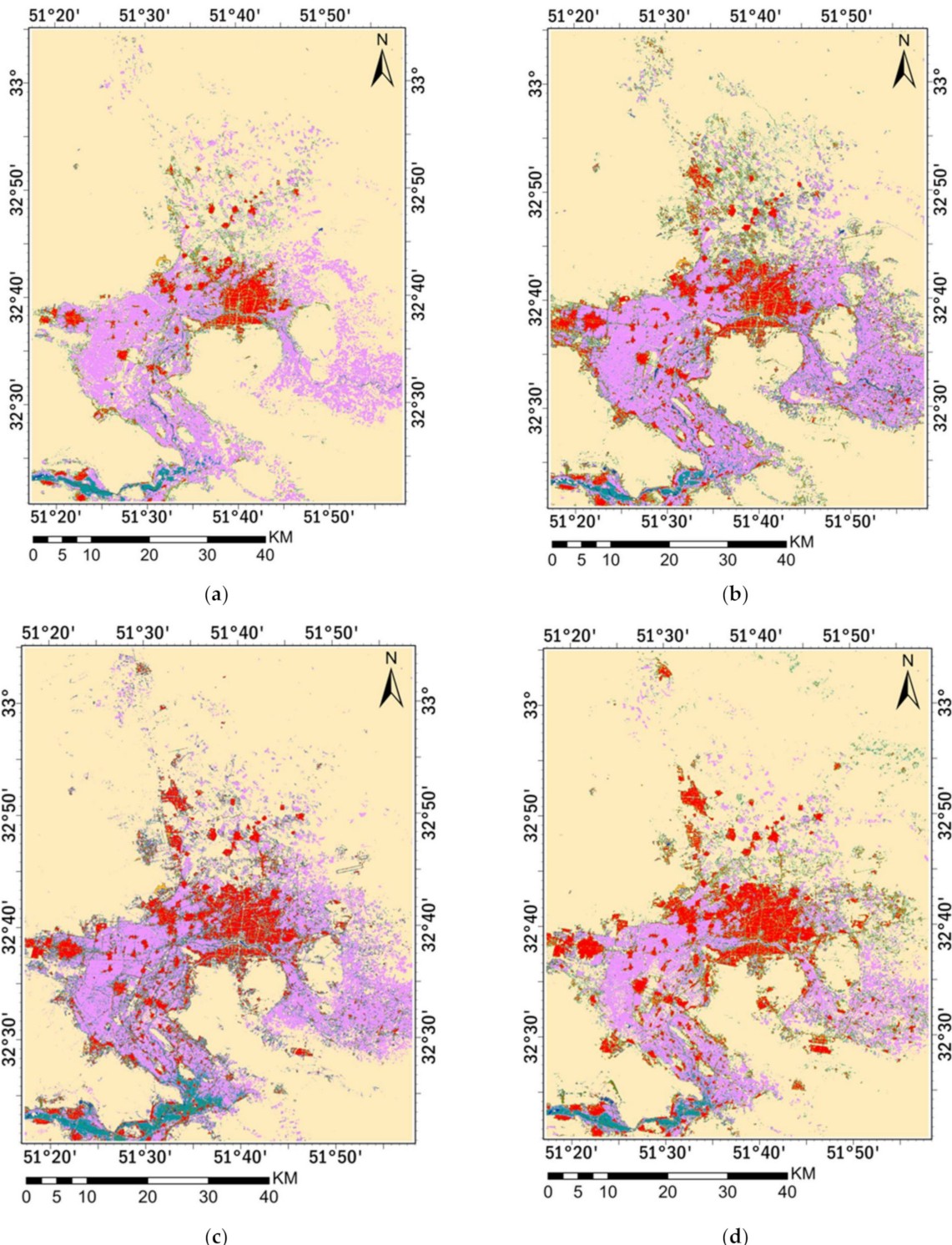

**Figure 5.** *Cont.*

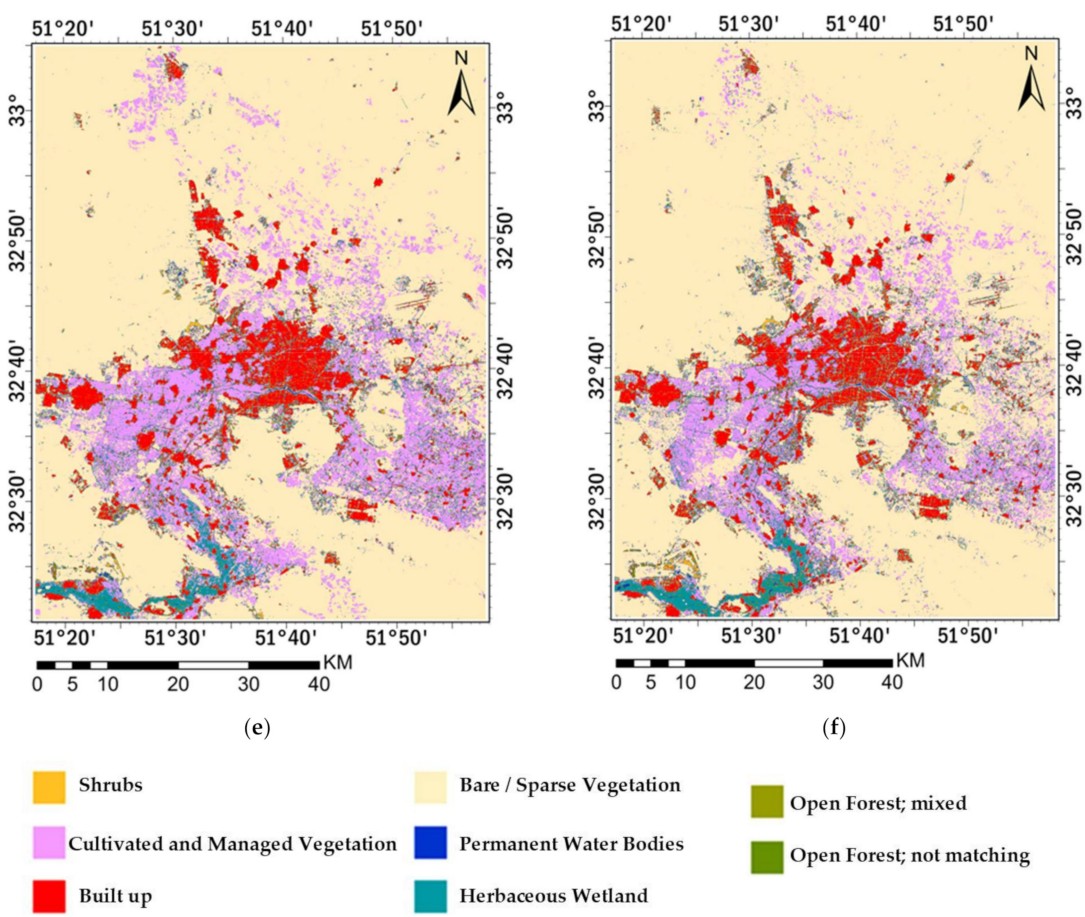

**Figure 5.** LULC map results: (**a**) 1985; (**b**) 1993; (**c**) 2000; (**d**) 2008; (**e**) 2013; (**f**) 2019.

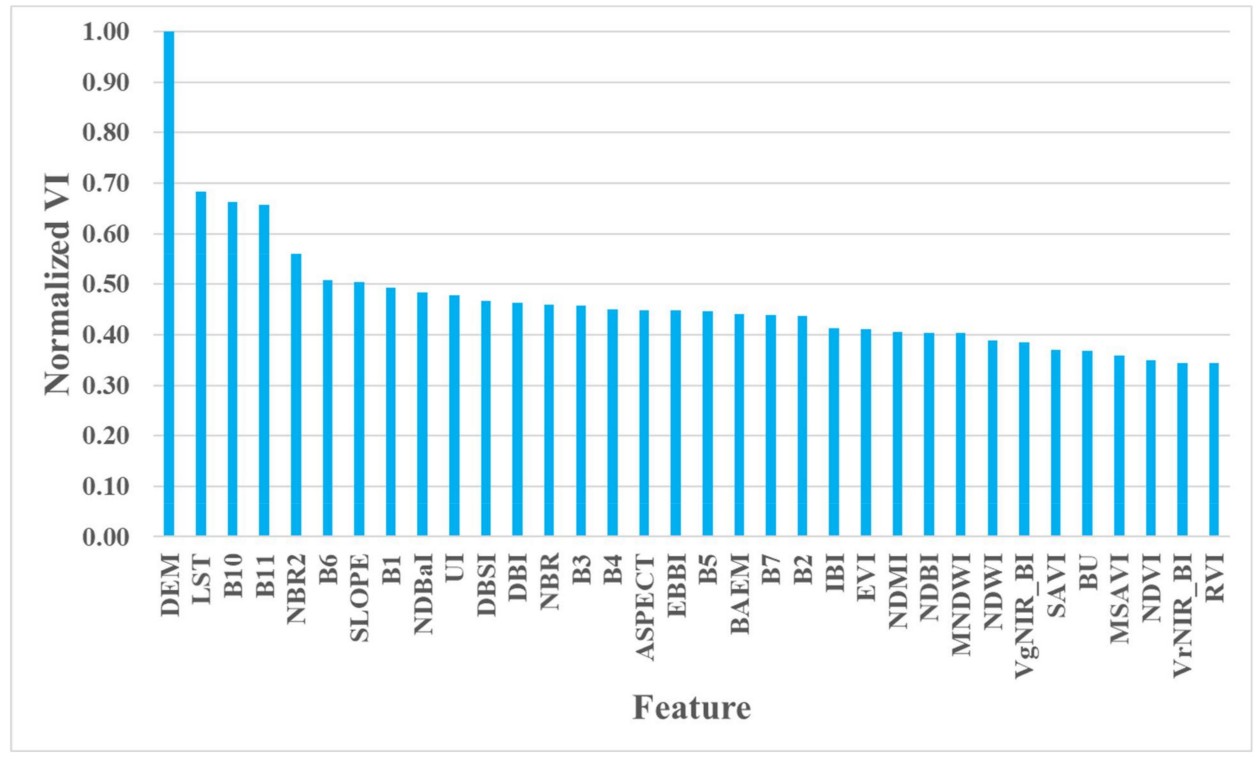

**Figure 6.** The average of normalized feature (variable) importance.

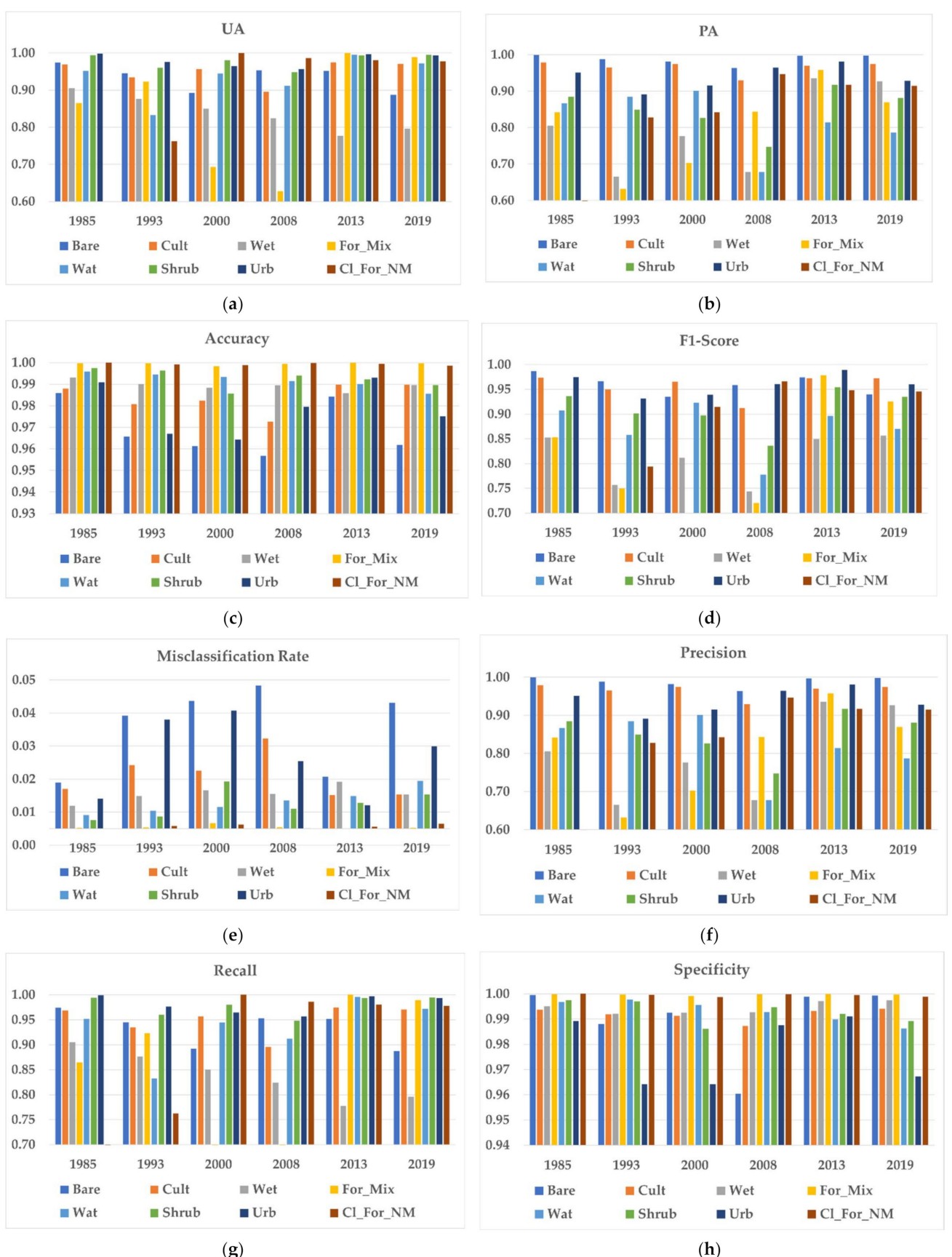

**Figure 7.** Estimated error matrix of sample counts with (**a**) user's accuracy; (**b**) producer's accuracy; (**c**) accuracy; (**d**) F1-score; (**e**) misclassification rate; (**f**) precision; (**g**) recall; (**h**) specificity.

**Table 6.** Estimated error matrix of sample counts.

| Year | OA | Kappa | Average Accuracy | Average F1-Score | Average Misclassification Rate | Average Precision | Average Recall | Average Specificity |
|---|---|---|---|---|---|---|---|---|
| 1985 | 97.55 | 0.96 | 0.99 | 0.93 | 0.01 | 0.90 | 0.95 | 1.00 |
| 1993 | 94.68 | 0.92 | 0.99 | 0.86 | 0.01 | 0.84 | 0.90 | 0.99 |
| 2000 | 93.64 | 0.92 | 0.98 | 0.89 | 0.02 | 0.87 | 0.91 | 0.99 |
| 2008 | 94.18 | 0.91 | 0.99 | 0.86 | 0.01 | 0.84 | 0.89 | 0.99 |
| 2013 | 96.73 | 0.96 | 0.99 | 0.95 | 0.01 | 0.94 | 0.96 | 1.00 |
| 2019 | 94.49 | 0.93 | 0.99 | 0.93 | 0.01 | 0.91 | 0.95 | 0.99 |

### 3.2. Map-to-Map Comparison

To investigate the performance of the proposed classification approach, we also compared the classification results to the global CGLCL map (Figure 8). Visual interpretation proved that the class of permanent water bodies was frequently classified as other classes (e.g., bare/sparse vegetation and cultivated), and the urban/built-up class could not successfully be extracted, especially along its borders. In addition, a statistical comparison was made between the proposed classifier and the global CGLCL map for visual comparison. According to the OA and Kappa, the proposed classifier outperforms the CGLCL map with 10.01% and 0.14.

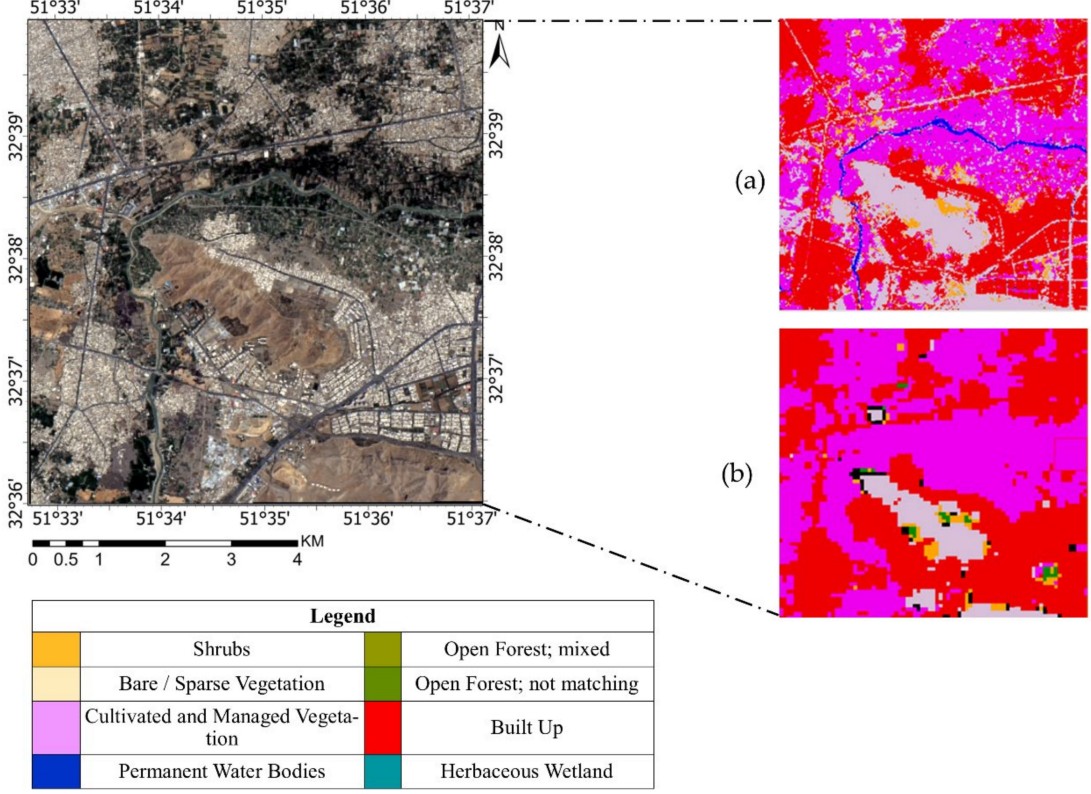

**Figure 8.** Comparison between our proposed method (**a**) and CGLCL map (**b**) in 2019.

### 3.3. LULC Change

Figure 9 displays the binary change detection maps where both unchanged and changed zones are pictured. Land cover changes in 1985–1993, 1993–2000, 2000–2008, 2008–2013, and 2013–2019 illustrate that 97,725.78, 105,689.43, 110,185.38, 111,393, and 103,546.98 ha of the study area were subjected to change. These changed areas are equivalent to 15.57, 16.84, 17.59, 17.75, and 16.5% of the study area for the corresponding periods. At

the class level, the results indicate that from 1985 to the 2019 situation, urban area, wetland, forest mixed, water, and shrubs increased by 167.97, 93.23, 151.68, 13.43, and 333.19%. In contrast, bare land and cultivated classes were reduced by 5.22 and 23.29%. Bare land and cultivated areas were mainly converted to urban class. The synergy between the change maps and GEE-derived indices/bands allowed the reconstruction of the annual land cover change maps from 1985 to 2019. Table 7 shows that, since 1985, tremendous land cover conversion, mainly from bare land and cultivated, witnessed urban expansion across the Isfahan region landscape. The study results illustrate extensive urban growth between 1985 and 2019, with a 167.97% growth (approximately 4.94% per year).

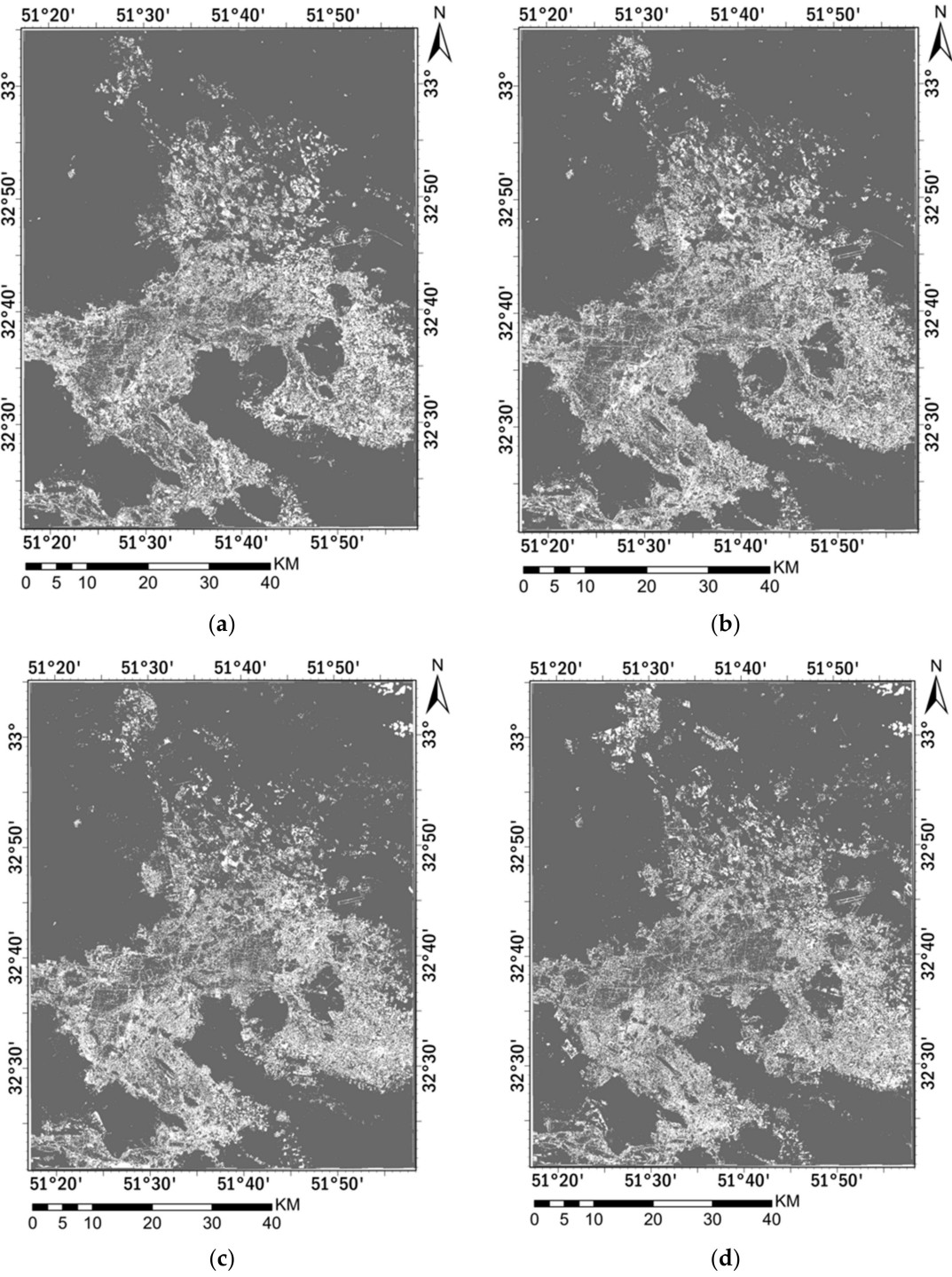

**Figure 9.** *Cont.*

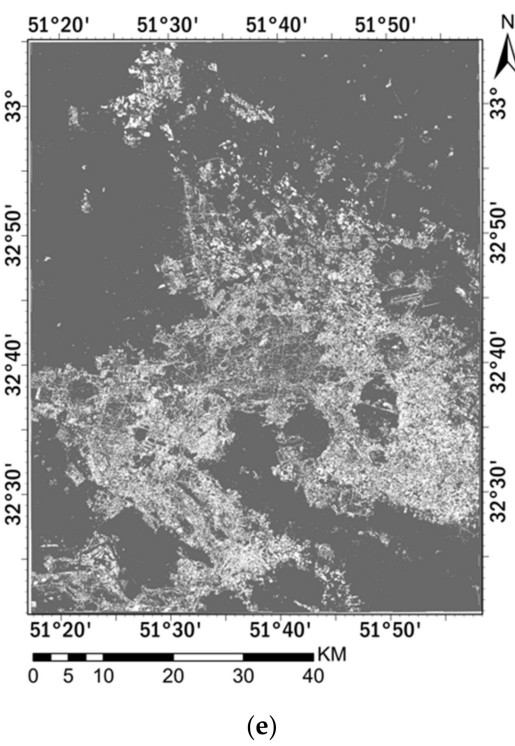

(**e**)

**Figure 9.** Change map between the time interval (the dark and light color indicate not-changed and changed areas, respectively.): (**a**) 1985–1993; (**b**) 1993–2000; (**c**) 2000–2008; (**d**) 2008–2013; (**e**) 2013–2019.

**Table 7.** LULC change (ha) from 1985 to 2019 based on classified Landsat data.

|  | 1985 | 1993 | 2000 | 2008 | 2013 | 2019 |
|---|---|---|---|---|---|---|
| Bare | 430,591.32 | 386,213.73 | 399,174.05 | 416,134.89 | 394,304.4 | 408,122.19 |
| Cult | 64,937.49 | 83,654.63 | 69,293.65 | 50,290.57 | 66,583.42 | 49,815.41 |
| Wet | 2418.93 | 2231.95 | 4689.74 | 2566.47 | 5261.77 | 4674.17 |
| For_Mix | 34.31 | 55.48 | 56.81 | 59.26 | 122.71 | 86.35 |
| Wat | 1450.97 | 2638.24 | 1176.24 | 1201.78 | 1203.99 | 1645.81 |
| Shrub | 1107.70 | 2280.15 | 1903.69 | 1608.15 | 3708.62 | 4798.48 |
| Urb | 18,595.7 | 41,538.94 | 43,015.1 | 45,981.29 | 47,970.07 | 49,830.55 |
| Op_For_NM | 0 | 169.83 | 90.65 | 342.51 | 306.66 | 534.18 |

Other classes, Shrub, Cult, Bare, Wat, Wet, For_Mix, and Op_For_NM, showed different increasing and decreasing behavior during some periods, which might be due to impacts of severe climate change and drought, rapid industrial growth, and sprawl-encouraging planning policies [35]. Table 7 and Figure 10 show the results. For example, the results indicate that vegetation classes have faced fluctuation during the past four decades. Meanwhile, the general tendency has inclined given worldwide climate change, which has caused severe drought conditions, leading to environmental degradation in this region.

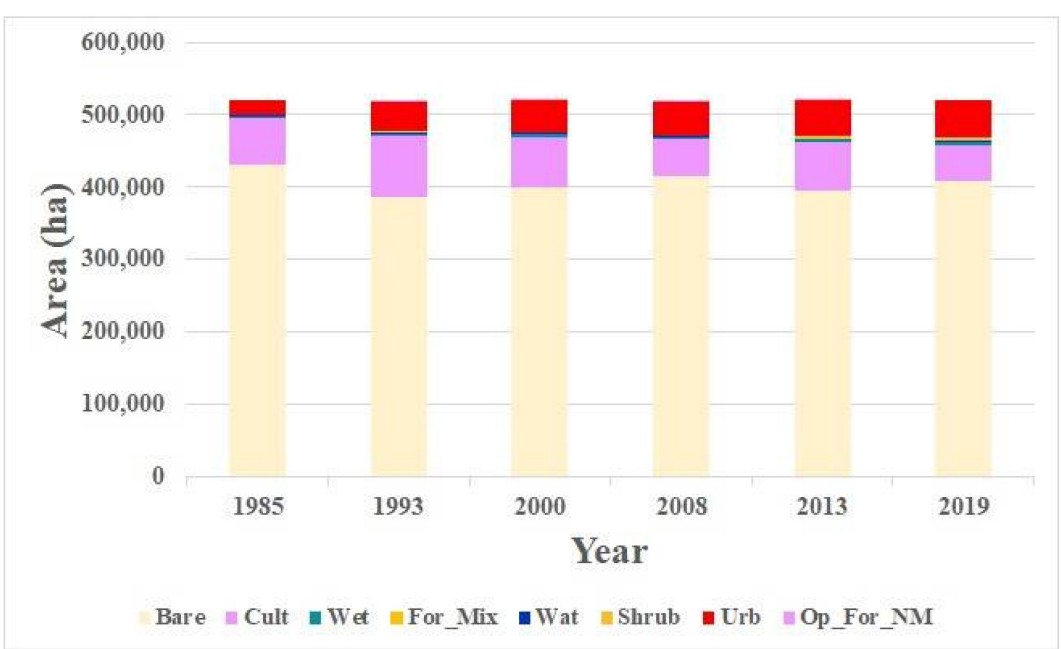

**Figure 10.** Dense annual LULC change from 1985 to 2019 based on classification results.

## 4. Discussion

A continuous increase in urbanization characterizes countries in the Middle East. LULC mapping is an effective tool for environmental monitoring and management. The LULC change occurs worldwide, but this phenomenon imitates great difficulties, mainly in metropolitan areas. These changes in the context of the study area, especially when it comes to urban growth and fluctuation in vegetation classes, are affected by economic (e.g., land price and land speculation), political (e.g., housing and tax policies, and civil war), demographic (e.g., population growth), environmental (e.g., enforced immigration and suburban environment), spatial planning (e.g., urban transport system), and climate change (e.g., severe drought) factors [36]. The case study presents urban environments with a rapid spread of informal settlements over the past four decades since 1985. We adopted the combination of several spectral indices and other auxiliary data. These results were valuable for a reliable land use classification in the study area. The combination of classifications in six proposed periods allowed continuous annual land cover classification from 1985 to 2019. It is necessary to mention that these datasets and the auxiliary variables have the exact spatial resolution, 30 m for Landsat 5 and 15 m for Landsat 7 and 8. In recent years, researchers used Sentinel-2 data to extract LULC [75]. Although Sentinel-2 data provides images with higher spatial resolution, the Landsat data might be a better source to investigate LULC change because of its capability to monitor Earth for more than five decades.

The results also show that a fair number of trees, one of the parameters of the RF classifier, is 50. Using a more significant number of trees would increase computations, but the results and accuracy would not improve. Comparing SR Landsat products with a pan-sharpened version of TOA Landsat product display, the impact of spatial resolution would be more effective than more accurate radiometric calibration. According to the results, DEM and LST are compelling features for LULC mapping and can impact attained accuracy. This study showed that LST, computed from thermal bands of Landsat images, can be an effective feature for discrimination between available LULC in the study area. The effectiveness of the LST feature might be because of different spectral signatures for each class in thermal bands. In addition, classes might have a relationship with spatial metrics such as height patterns, which helped increase the accuracy of the final classification. The results also show that global LULC maps are unsuitable for accurate mapping of LULC change maps, and introducing new algorithms and training data over local areas is

essential. The inter-annual change analysis revealed that abrupt change to urban structures was observed from 1985 to 1993 (coincided with the baby boom and postwar economic growth, with an annual averaging rate of 15.42%. During that period, the built-up area extended from 18,595.7 to 41,538.94 hectares. From 1995 to 2019, a gradual increasing urban growth shows stability in urban policies and development plans, with an annual average rate ranging between 0.51% and 0.86%. This gradual increase is related to Iran's economic, environmental, and spatial planning conditions [36]. According to [36], the growth rate of the core city of Isfahan would decrease, and other sub-cities would face a more significant growth rate.

This research shows that 918.67 ha on average between 1985 and 2019 has changed to urban area class yearly over the study area. It shows that many factors, such as economic reasons, have accelerated urban growth, and it puts noticeable pressures on human welfare and the natural environment. This urban growth also increases the rate of rural-urban migration, which causes environmental degradation, especially in metropolitan areas such as Isfahan. The results also show that vegetation classes have faced fluctuation during this time interval, but the general trend was inclined to decrease. This phenomenon is under the impact of several reasons, such as climate change, which caused severe drought conditions. To reduce the long-term consequences of this LULC change, it is necessary to provide an accurate LULC map and build a practical scenario.

## 5. Conclusions

This study presents a developed method to map LULC with high accuracy at the scale of 30 m for Landsat 5 and 15 m for Landsat 7 and 8 based on GEE's cloud-based platform. For this purpose, we used the RF algorithm, which is one of the most robust classifiers. We checked a different number of trees and input RF parameters, and the results show that 50 is a suitable value. The results also showed that although using SR products instead of TOA Landsat products would result in a more accurate classification map, it is not correct if a pan-sharpening method is applied. The proposed method's OA, Kappa, and F1-score were 0.422%, 0.006, and 0.015 better than the similar approach used for the SR version of Landsat data in 2019. Two spectral-temporal features (7, 8, and 9 raw bands for Landsat 5, 7, and 8 and 25 indices and auxiliary data for all Landsat collections) characterize the LULC. In addition, we used DEM and extracted data from DEM (slope and aspect). According to the results, the LST and DEM features are essential in Landsat image classification. Although these features are crucial in LULC mapping, they are not used, or just one of them is used in similar research. We used a CGLCL map and high-resolution Google Earth images to provide reliable training class labels. The OA, Kappa values, and F1-score are 93.64–97.55%, 0.91–0.96, and 0.86–0.95, respectively. Independent and map-to-map comparison assessment approaches verified the final classification results. The results showed clearly that the proposed method outperforms the CGLCL map. The proposed method's OA, Kappa, and F1-score were 10%, 0.13, and 0.5 better than the CGLCL map in 2019.

This paper proposed a method based on the GEE cloud-based platform to accurately and timely map the change of LULC from historical Landsat data. This method can be quickly and easily applied to other regions of interest for LULC mapping. As the code is available publicly, future improvements to this methodology may be implemented based on user feedback. We recommend that the importance of the LST and DEM features can be checked in other available satellite image data for LULC mapping. In addition, the LULC map would present a clear understanding of the spatial, environmental, and socioeconomic pattern of changes and allows the governments to minimize the cost and negative impacts of these changes. The result of this study can also be used for land-use planning and predicting urban expansion. In this study, the impact of the pan-sharpening methods was not investigated. When we did this study, 10-m landcover such as Esri 10-Meter Land Cover (10-class) and 10 m Annual Land Use Land Cover (9-class) datasets were not released.

Therefore, we did not compare our results with these two datasets. We implemented our methodology in the GEE platform.

Moreover, these two global LULC datasets are not available in the GEE. The only global land cover map at 10 m resolution in GEE is ESA WorldCover 10 m v100. However, this dataset is only available for 2020 and later. For future study, the effectiveness of datasets such as Landsat-9 and the combination of Landsat and Sentinel for LULC change can be investigated. More features (e.g., textural features such as GLCM) can also be added to improve the accuracy of LULC change analysis. The results can also be compared to the global LULC maps mentioned earlier.

**Author Contributions:** Conceptualization, S.H., H.R.-D. and S.A.; methodology, S.H., S.A. and M.S.; software, M.S.; validation, S.A.; writing—original draft preparation, S.A.; writing—review and editing, S.H., H.R.-D. and S.A.; visualization, S.A. and M.S.; supervision, S.H. and H.R.-D.; project administration, S.H. All authors have read and agreed to the published version of the manuscript.

**Funding:** Hamidreza Rabiei-Dastjerdi is a Marie Skłodowska-Curie Career-FIT Fellow at the UCD School of Computer Science and CeADAR (Ireland's National Center for Applied Data Analytics and AI). Career-FIT has received funding from the European Union's Horizon 2020 research and innovation program under the Marie Skłodowska-Curie Grant Agreement No. 713654.

**Data Availability Statement:** The GEE repository and the view from results are publicly available at: https://code.earthengine.google.com/?accept_repo=users/MohsenSaber/LCC_Isfahan (accessed on 26 April 2022) and https://mohsensaber.users.earthengine.app/view/isfahan-lcc (accessed on 26 April 2022).

**Acknowledgments:** The authors would like to thank Sofia Ermida for the LST retrieval code (https://code.earthengine.google.com/?accept_repo=users/sofiaermida/landsat_smw_lst) (accessed on 26 April 2022). The authors would like to thank net friends from gis.stackexchange.com for the GEE operation's generously professional instruction.

**Conflicts of Interest:** The authors declare no conflict of interest.

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
