# Peer review of "Urban Land Use and Land Cover Change Analysis Using Random Forest Classification of Landsat Time Series"

_remotesensing, doi:10.3390/rs14112654_

Round 1

Reviewer 1 Report

Reviewer’s Report on the manuscript entitled:

Urban Land Use and Land Cover Change Analysis Using Random Forest Classification of Landsat Time-Series

The authors analyzed the urban land use and land cover (LULC) change in Isfahan using the random forest method through GEE. The topic and results are interesting and useful. Please find my comments below for further improvement.

The abstract needs improvement.

Line 14. Please add a sentence to talk about the importance of LULC to better attract readers to the article.

Please remove the hyphen in “time-series” everywhere in the manuscript.

Please write LANDSAT as Landsat.

Line 17. Please replace “time periods” with “period”. Also please remove “In this paper,”

Line 27. Replace LST with “land surface temperature” and DEM with “digital elevation model”.

Line 67. Please also add Deep Transfer Learning (DTL) and add the following articles:

https://doi.org/10.3390/s21238083

https://doi.org/10.3390/rs12010086

Line 116. Please define CGLCL here. Please also check the abbreviations to ensure they are all defined the first time they appeared (in addition to the Abstract).

Line 118. Please add a paragraph here to describe how the rest of the paper is organized. For example, "The rest of the paper is organized as follows. Section 2 describes the study region, datasets, and methods. Section 3 demonstrates the results, …”

Figure 6. Please remove “(a)”. Also, your figures can have the same font (Arial recommended). Figure 6 looks like Time New Roman while figure 5 looks like it is Calibri. I suggest using the same font for all figures. Consistency will add value to the manuscript. Also, please ensure the resolutions of the figures are at least 300 dpi.

Line 381. “show” not “shows”

Figure 9. What do the white color areas in these panels represent? Are they LULC change? Please describe in the figure caption.

Table 7 caption. Is the period 1990 to 2015 or 1985 to 2019? I see the years 1985, 1993, 2000, 2008, 2013, and 2019.

Line 398. Please remove one set of parentheses.

Line 419. Please re-write this sentence. It is grammatically incorrect.  

Please also mention the use of Sentinel 2 imagery for  urban LULC classification in the Discussion part:

https://doi.org/10.1007/s11042-021-10991-0

Please also briefly describe the limitations of the study, computational complexity, and future direction in the Conclusion part.

Please follow the MDPI guideline for formatting all the references.

Thank you for your contribution

Regards,

Author Response

The authors are very thankful to the reviewer for the suggestions to improve the quality and presentation of the manuscript. Following the comments made, the manuscript has been duly revised. The reviewer’s concerns are addressed in the following item-by-item form with sincere thanks from the authors. All revisions made to manuscripts are marked up using "Track Change" function of MS Word.

  1. Comment “The abstract needs improvement.”

Response: Thank you for this constructive suggestion. We revised the abstract section as you asked.

  1. Comment “Line 14. Please add a sentence to talk about the importance of LULC to better attract readers to the article”

Response: We revised the abstract section and described the importance of LULC in this section.

  1. Comment “Please remove the hyphen in “time-series” everywhere in the manuscript.”

Response: We edited the paper for such mistakes, and the changes are highlighted in the last version of the paper.

  1. Comment “Please write LANDSAT as Landsat.”

Response: Thank you for your scrutiny. We did that.

  1. Comment “Line 17. Please replace ‘time periods’ with ‘period’. Also please remove ‘In this paper,’”

Response: We replaced this word as you asked.

  1. Comment “Line 27. Replace LST with ‘land surface temperature’ and DEM with ‘digital elevation model’.”

Response: Thank you for this constructive suggestion. We revised the paper as you asked.

  1. Comment “Line 67. Please also add Deep Transfer Learning (DTL) and add the following articles: https://doi.org/10.3390/s21238083 & https://doi.org/10.3390/rs12010086”

Response: Thank you for the insightful reference. We added this reference to the latest paper version and edited the text. Now the text is smoother for the reader.

  1. Comment “Line 116. Please define CGLCL here. Please also check the abbreviations to ensure they are all defined the first time they appeared (in addition to the Abstract).”

Response: We thank the reviewer for this observation; the problem has been fixed.

  1. Comment “Line 118. Please add a paragraph here to describe how the rest of the paper is organized. For example, ‘The rest of the paper is organized as follows. Section 2 describes the study region, datasets, and methods. Section 3 demonstrates the results, …’”

Response: This part is added to the latest version of the paper.

  1. Comment “Figure 6. Please remove “(a)”. Also, your figures can have the same font (Arial recommended). Figure 6 looks like Time New Roman while figure 5 looks like it is Calibri. I suggest using the same font for all figures. Consistency will add value to the manuscript. Also, please ensure the resolutions of the figures are at least 300 dpi.”

Response: We modified the font of the figures’ caption following the available sample file in this journal.

  1. Comment “Line 381. “show” not “shows””

Response: Thank you for the insightful comments. We edited the paper for such mistakes.

  1. Comment “Figure 9. What do the white color areas in these panels represent? Are they LULC change? Please describe in the figure caption.”

Response: We added a caption to Fig. 9.

  1. Comment “Table 7 caption. Is the period 1990 to 2015 or 1985 to 2019? I see the years 1985, 1993, 2000, 2008, 2013, and 2019.”

Response: We thank the reviewer for pointing out this problem.

  1. Comment “Line 398. Please remove one set of parentheses.”

Response: We revised the paper for such problems.

  1. Comment “Line 419. Please re-write this sentence. It is grammatically incorrect.”

Response: Thank you for your scrutiny.

  1. Comment “Please also mention the use of Sentinel 2 imagery for urban LULC classification in the Discussion part: https://doi.org/10.1007/s11042-021-10991-0”

Response: We discussed Sentinel-2 data in the latest version of the paper.

  1. Comment “Please also briefly describe the study’s limitations, computational complexity, and future direction in the Conclusion part.”

Response: We thank the reviewer for pointing out this problem. We described these points in the conclusion section.

  1. Comment “Please follow the MDPI guideline for formatting all the references.”

Response: We revised the paper following the MDPI guideline for references’ formatting.

Reviewer 2 Report

The application of pan-sharpening has no effect on the Landsat 5 collection data (as there is no PAN band in this collection) and this is not made clear in the article. It must be explained.

Author Response

The authors would like to thank the reviewer for his/her precious time and invaluable comments. We have carefully addressed all the comments. The manuscript has been duly revised, and the concerns of the reviewer are addressed in the following item-by-item form. All revisions made to manuscripts are marked up using "Track Change" function of MS Word.

  1. Comment “The application of pan-sharpening has no effect on the Landsat 5 collection data (as there is no PAN band in this collection) and this is not made clear in the article. It must be explained.”

Response: Thank you for the constructive comments. In the latest version of the paper, we explained that pan sharpening is only applied on Landsat 7 and 8.

Reviewer 3 Report

The authors implemented the classification and change analysis of urban land use and land cover based on Google Engine cloud platform using random forest algorithm. In this process, the pan-sharpening algorithm and Variable importance (VI) are especially considered.  The technical route of the study is clear and reasonable, and the following suggestions are available for reference.
1. In the abstract section, the author should have 1-2 sentences about the background of the study.

2. 2.4.1 Random forest algorithm is a general and mature algorithm in the field of remote sensing. For its algorithmic process, except for the redesigned points, it should be simplified.

3. 261 lines, the Variable importance (VI) of features is calculated, which itself is very necessary, can the authors add the description of how to eliminate redundant features. In addition, the authors mention DEM and LST, which can be seen from the results to have a significant impact on the classification results. Is it possible to add some discussion in the discussion section, in addition, it is suggested to add the topographic base map in Figure1.

4. Figure3 suggests redrawing to improve the resolution.

5. The resolution of the classification results is 30m, and the comparison data CGLCL is 100m, so the direct comparison is not comparable. How did the authors handle this?

6. Please check all figures and tables  to make sure there are horizontal and vertical coordinates with names and units, etc.

Author Response

The authors are very thankful to the reviewer for the suggestions to further improve the quality and presentation of the manuscript. Following the comments made, the manuscript has been duly revised. The reviewer’s concerns are addressed in the following item-by-item form with sincere thanks from the authors. All revisions made to manuscripts are marked up using "Track Change" function of MS Word.

  1. Comment “In the abstract section, the author should have 1-2 sentences about the background of the study.”

Response: We revised the abstract of the paper.

  1. Comment “2.4.1 Random Forest algorithm is a general and mature algorithm in the field of remote sensing. For its algorithmic process, except for the redesigned points, it should be simplified.”

Response: Thank you for the constructive comments. We revised this section.

  1. Comment “261 lines, the Variable importance (VI) of features is calculated, which itself is very necessary, can the authors add the description of how to eliminate redundant features. In addition, the authors mention DEM and LST, which can be seen from the results to have a significant impact on the classification results. Is it possible to add some discussion in the discussion section, in addition, it is suggested to add the topographic base map in Figure 1.”

Response: More details about VI are added to the RF classifier section. We discussed DEM and LST in the discussion section. Figure 1 is modified as you asked.

  1. Comment “Figure3 suggests redrawing to improve the resolution.”

Response: Thank you for this suggestion. It is done.

  1. Comment “The resolution of the classification results is 30m, and the comparison data CGLCL is 100m, so the direct comparison is not comparable. How did the authors handle this?”

Response:

When we did this study, 10-meter landcover like Esri 10-Meter Land Cover (10-class) and 10m Annual Land Use Land Cover (9-class) datasets were not released. Therefore, we did not compare our results with these two datasets. We implemented our methodology in the GEE platform. In the GEE, these two global LULC datasets are not available. The only global land cover map at 10 m resolution in GEE is ESA WorldCover 10m v100. However, this dataset is only available for 2020 and later. We also added it as a future direction in the conclusion part. We compared our results with Copernicus global land cover (CGLC) map. CGLC map has a different spatial resolution compared to our LULC map.

To solve this problem, we used image.reproject command to transfer CGLC map into the spatial resolution of our results. The inputs for this command are image, coordinate system, and scale, respectively. More information can be found in the following link:

https://developers.google.com/earth-engine/apidocs/ee-image-reproject

  1. Comment “Please check all figures and tables to make sure there are horizontal and vertical coordinates with names and units, etc.”

Response: We thank the reviewer for this observation; We checked the paper for such mistakes.